# RealEra: Semantic-level Concept Erasure via Neighbor-Concept Mining

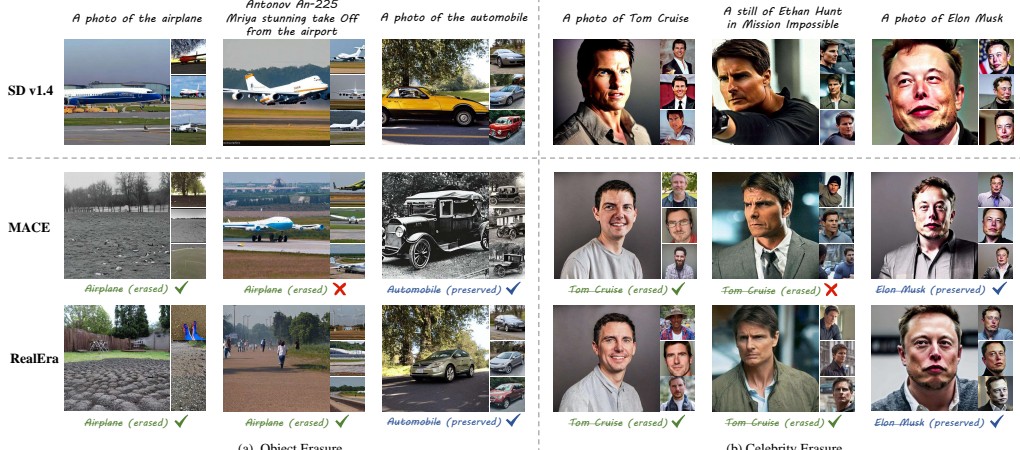

Figure 1: For text inputs closely associated in semantics but not explicitly containing the erasure concept, previous methods still generate objects of erasure concept, defined as the *concept residue* issue. For example, when it comes to concept of "airplane", if we input "Antonov An-225 Mriya stunning take off from the airport", which is a specific name of aircraft, previous MACE method still generates an image of airplane. While our RealEra method shows the *real erasure* on airplane, showing the trade-off between efficacy and specificity.

## ABSTRACT

The remarkable development of text-to-image generation models has raised notable security concerns, such as the infringement of portrait rights and the generation of inappropriate content. Concept erasure has been proposed to remove the model's knowledge about protected and inappropriate concepts. Although many methods have tried to balance the efficacy (erasing target concepts) and specificity (retaining irrelevant concepts), they can still generate abundant erasure concepts under the steering of semantically related inputs. In this work, we propose RealEra to address this "concept residue" issue. Specifically, we first introduce the mechanism of neighbor-concept mining, digging out the associated concepts by adding random perturbation into the embedding of erasure concept, thus expanding the erasing range and eliminating the generations even through associated concept inputs. Furthermore, to mitigate the negative impact on the generation of irrelevant concepts caused by the expansion of erasure scope, RealEra preserves the specificity through the beyond-concept regularization. This makes irrelevant concepts maintain their corresponding spatial position, thereby preserving their normal generation performance. We also employ the closed-form solution to optimize weights of U-Net for the cross-attention alignment, as well as the prediction noise alignment with the LoRA module. Extensive experiments on multiple benchmarks demonstrate that RealEra outperforms previous concept erasing methods in terms of superior erasing efficacy, specificity, and generality.

## 1   INTRODUCTION

In recent years, the surge of generative artificial intelligence (GAI) has brought historic opportunities for the development of various fields, especially in text-to-image generation (T2I) (Nichol et al., 2022; Ramesh et al., 2022; Rombach et al., 2022; Saharia et al., 2022b). The T2I diffusion models have produced images of remarkable quality, gratitude to its training on large-scale Internet datasets. However, these unfiltered large-scale datasets contains abundance of Not-Safe-For-Work (NSFW) content (Hunter, 2023; Zhang et al., 2023), as well as images involving intellectual property (Jiang et al., 2023; Roose, 2022; Setty, 2023) or portrait rights (Somepalli et al., 2023). Diffusion models even learn and memorize these concepts (Carlini et al., 2023; Kumari et al., 2023), making it easy for users to generate harmful or infringing content, and leading to the spread of disinformation and greater harm to the society.

To address this security issue, researchers have designed several safety mechanisms for T2I diffusion models. An intuitive solution is to retrain the model using the filtered images (Rombach, 2022), which whereas not only requires expensive computational costs but also leads to a decrease in generation quality. In addition, the NSFW safety checker which tries to filter out the inappropriate results after generation (Rando et al., 2022), while the classifier-free guidance aims at eliminating the concept generation in inference phase (Schramowski et al., 2023). However, they can be easily circumvented by malicious users due to the open-source model parameters and code.

Recently, some methods propose to erase these concepts by fine-tuning T2I diffusion models (Gandikota et al., 2023; 2024; Zhang et al., 2024; Kumari et al., 2023; Heng & Soh, 2024; Lu et al., 2024; Lyu et al., 2024). Specifically, for text inputs containing inappropriate concepts, they adjust the internal parameters of generation model through fine-tuning, so that the generated content no longer contains these concepts. Previous work has reached a consensus on the need to solve the trade-off between efficacy and specificity in concept erasure. Given a text input containing the erasure concept, efficacy means that the model outputs irrelevant content while maintaining the overall naturalness. While specificity implies that if the text input has no relation to the erasure concept, the output should remain identical to the original model before erasure.

Despite their appealing performance, they fail to produce surprising results when encountering implicitly associated input concept. As shown in Figure 1, for text inputs closely associated in semantics but not explicitly containing the erasure concept, previous methods still generate objects of erasure concept. For example, when it comes to concept of "Tom Cruise", if we input "A still of Mission Impossible", which is Tom Cruise's most iconic work, previous method can still generate a portrait of Tom Cruise. Note that "Mission Impossible" is a concept closely associated with "Tom Cruise", which whereas doesn't explicitly include "Tom Cruise". We define this as the *concept residue* issue, i.e., erasure concept still exists in some implicitly associated concepts. This fails to be tackled by previous methods, which to some extent considered as the word-level erasure, and thus becomes the motivation of this work.

In this paper, we propose a novel concept erasure framework, named RealEra, which prevents the diffusion model from regenerating erasure concepts with semantically related inputs. Specifically, we first mine associated concepts by randomly sampling within the vicinity space of erasure concept. By introducing the stochasticity into erasure concept's embedding and shifting it to an associated concept, RealEra steers the associated concept to the anchor concept. Meanwhile, erasing one concept from diffusion models should prevent the catastrophic forgetting of others, whereas simply suppressing the generation of erasure concept leads to severe concept erosion. To maintain specificity preservation, we introduce the beyond-concept regularization, which turns the erasure concept into concepts that are far away and irrelevant by sampling perturbation outside the neighborhood range. This makes irrelevant concepts maintain their corresponding spatial position, thereby preserving their normal generation performance. Subsequently, we employ the closed-form solution to optimize weights of U-Net for the cross-attention alignment, as well as the prediction noise alignment with the LoRA module. RealEra achieves superior performance in both erasing assigned concepts, and preserving the generation ability of other unrelated concepts.

Our contributions can be summarized as follows:

- We present a novel concept erasure framework RealEra to solve the concept residue issue because of associated concept input, which aims at steering the associated concept to the anchor concept by mechanism of neighbor-concept mining.

- RealEra also employs specificity preservation with beyond-concept regularization, which compensates for the negative impact of erasing associated concepts on unrelated concepts.

- Extensive experiments demonstrate that the our method greatly boosts the effectiveness of concept erasure, especially for the implicitly associated concept inputs.

## 2 RELATED WORK

### 2.1 TEXT-TO-IMAGE GENERATION

Training on large-scale datasets, the text-to-image (T2I) generation models have achieved great success recently. T2I generation involves creating visual images from textual prompts, which has made significant advances with diffusion models. Various methods have been developed to achieve high-resolution text-to-image generations. As a pioneer work, GLIDE (Nichol et al., 2022) trains a 3.5B text-conditional diffusion model at a $64 \times 64$ resolution, as well as a 1.5B parameter text-conditional up-sampling diffusion model to increase the resolution to $256 \times 256$. DALL-E 2 (Ramesh, 2023) proposes transforming a CLIP (Radford et al., 2021) text embedding into a CLIP image embedding with a prior model, and then decoding this image embedding into the image. Imagen (Saharia et al., 2022a) adopts a cascaded diffusion model and T5, a large pretrained language model, as text Encoder to generate images. Stable Diffusion (SD) (Rombach et al., 2022) is built on the latent diffusion model, which operates on the latent space instead of pixel space, enabling SD to generate high-resolution images. SD v1.x employs 123.65M CLIP as text encoder and trains at different steps on the laion-improved-aesthetics or laion-aesthetics v2 5+ datasets. SD v2.x uses laion-aesthetics v2 4.5+ datasets, a larger dataset and 354.03M OpenCLIP (Cherti et al., 2023), a more powerful CLIP text encoder.

### 2.2 CONCEPT ERASURE IN DIFFUSION MODELS

T2I models (Nichol et al., 2022; Ramesh et al., 2022; Rombach et al., 2022; Saharia et al., 2022b) are mostly trained on large-scale web-scraped datasets, such as LAION-5B (Schuhmann et al., 2022). Unfiltered datasets can cause T2I models to learn and generate a series of inappropriate content that violates copyright and privacy. To alleviate this concern, many studies explore and devising various solutions: training datasets filtering (Rombach, 2022), post-generation content filtering (Rando et al., 2022), classifier-free guidance (Schramowski et al., 2023), and fine-tuning pretrained models (Gandikota et al., 2023; 2024; Zhang et al., 2024; Kumari et al., 2023; Heng & Soh, 2024; Lu et al., 2024; Lyu et al., 2024). The Stable Diffusion 2.0 (Rombach, 2022) applies an NSFW detector to filter inappropriate content from the training dataset, but this leads to high retraining costs and generation quality decrease. Post-generation content filtering adopts the safety checker to filter out NSFW content, but this can easily be disabled by users. SLD (Schramowski et al., 2023) suppresses the generation of inappropriate content during the inference process with negative guidance, based on the classifier-free guidance.

Recently, some researches erase inappropriate concepts by fine-tuning the parameters of the T2I models. ESD (Gandikota et al., 2023) fine-tunes the pretrained model by guiding the model output away from the erasure concept with a negative conditioned score, so that the model learns from its own knowledge to steer the diffusion process away from the undesired concept. FMN (Zhang et al., 2024) utilizes attention re-steering to fine-tune UNet to minimize each of the intermediate attention maps associated with the erasure concepts. AC (Kumari et al., 2023) fine-tunes the model to match the prediction noise between the erasure concepts and corresponding anchor concepts, so that steering the erasure concepts towards anchor concepts. SA (Heng & Soh, 2024) incorporates EWC and generative replay to forget the erasure concept and remember the retention concepts, respectively. UCE (Gandikota et al., 2024) employs a closed-form solution to optimize the cross-attention weights of pretrained models, thereby mapping erasure concepts to anchor concepts. MACE (Lu et al., 2024) trains a separate LoRA module for each erasure concept by combining a closed-form method and minimizing activation values of erasure concept, and fuses multiple LoRA modules to achieve mass concepts erasure. SPM (Lyu et al., 2024) proposes to train a lightweight adapter for each erasure

concept, adopts a latent anchoring strategy to re-weight preserve loss based on semantic similarity, and utilizes an facilitated transport mechanism to regulate the multiple concepts erasure. However, these methods have not focused on the erasure performance of implicitly associated concepts while ensuring overall erasure performance.

## 3 PRELIMINARIES

### 3.1 LATENT DIFFUSION MODEL

To enhance the generative efficiency of the diffusion model, Latent Diffusion Model (LDM) (Rombach et al., 2022) proposes to shift the diffusion process from the pixel space of the images to the low-dimensional latent space, so it needs to train a VAE (Kingma, 2013) model to encode and decode images. In addition, in order to achieve conditional generation, text or image is converted into condition embedding and fed into the diffusion model, then the diffusion process is controlled conditionally by the attention mechanism in U-Net. Given a user's input prompt, it is first encoded into text embedding by the text encoder. The text embedding will then be projected into $K$ and $V$ vectors respectively by the projection matrix $W_K$ and $W_V$. Then, the $K$ vectors dot-product with $Q$ vectors from the noisy image to get the attention maps. The image-text fusion feature is obtained by multiplying the attention maps and $V$ vectors, then the final predicted noise is derived from the subsequent network structure of U-Net. The final training optimization objective is:

$$\mathcal{L}_{LDM} = \mathbb{E}_{z_t, t, c, \epsilon \sim \mathcal{N}(0,1)} \left[ \| \epsilon - \epsilon_\theta \left( z_t, t, c \right) \|_2^2 \right], \tag{1}$$

where $z_t$ represents the latent space variable of image $x$ through the VAE, $c$ represents the multimodal condition inputs such as text or image, connected to the diffusion model through the cross-attention mechanism.

### 3.2 LOW-RANK ADAPTATION

To reduce the cost of downstream transfer learning for large-scale models, the concept of parameter-efficient fine-tuning (PEFT) (Mangrulkar et al., 2022) has been introduced. Low-Rank Adaptation (LoRA) (Hu et al., 2021), as a structure in PEFT, enhances parameter efficiency by freezing the pre-trained weight matrices and integrating additional trainable low-rank matrices within the network. This method is based on the observation that pre-trained models exhibit low "intrinsic dimension". Given the pretrained weights matrix of the diffusion model $W \in \mathbb{R}^{m \times n}$, LoRA constrain its update with a low-rank decomposition $W' = W + BA$, where $B \in \mathbb{R}^{m \times r}$ and $A \in \mathbb{R}^{r \times n}$, and satisfying $r \ll \min(n, m)$. In training phase, only $A$ and $B$ are trainable and receive gradient updates, while $W'$ is frozen. $W'$ and $BA$ multiply the same input and sum them to output. Thus, as for input $x$ and output $h$:

$$h = W'x = Wx + BAx, \tag{2}$$

where LoRA adopts a random Gaussian initialization for $A$ and zero for $B$.

## 4 METHOD

### 4.1 EFFICACY ERASURE WITH NEIGHBOR-CONCEPT MINING

The previous methods mainly focus on mapping the erasure concepts to the anchor one. However, as for erasure concept, there are still multiple related concepts within its neighborhood that can easily condition the diffusion model to generate erasure concepts, e.g., "airport" and "An-225 Mriya" for "airplane", and "Mission Impossible" for "Tom Cruise". Therefore, to prevent the model from generating erasure concept through these associated concepts, we propose the mechanism of Neighbor-Concept Mining. Specifically, when fine-tuning the model, we add random perturbations to the input embedding of erasure concept, shifting them towards associated concepts in the adjacent semantic space. We fine-tune the diffusion model by mapping both these mined-out concepts and erasure concept to the anchor concept. Regarding the addition of stochasticity introduced by random sampling, we design the following scheme: suppose we have the prompt $p_c$ corresponding to the erasure concept $c$, whose corresponding embedding is $e$, and the defined perturbation as $\eta$. For

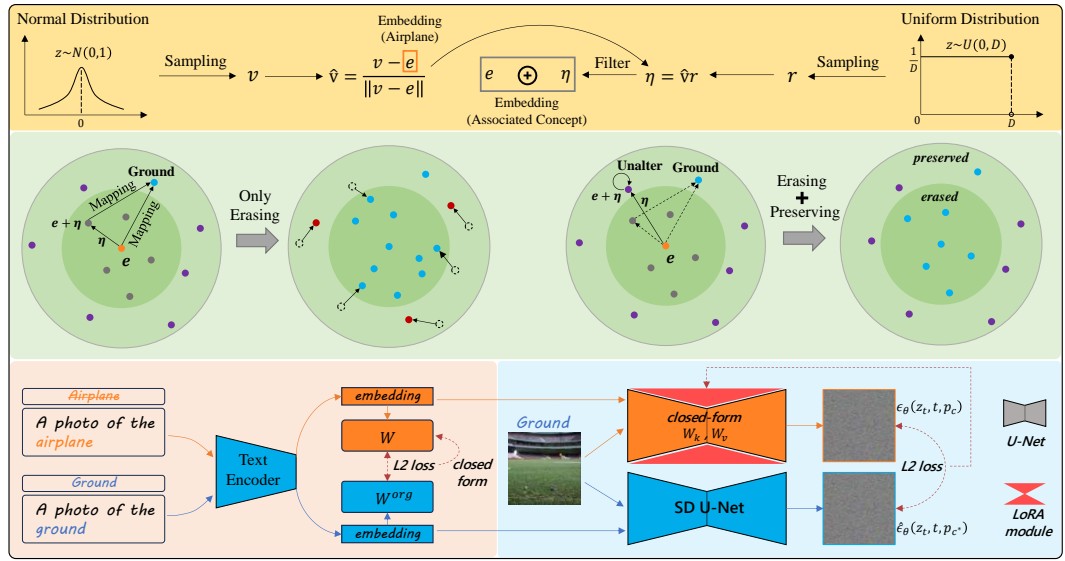

Figure 2: The overall pipeline of the proposed RealEra method. We mine and erase the associated concepts in the neighborhood of the erasure concepts, and to remain the mapping relationship of other unrelated concepts, we introduce additional beyond-concept regularization to preserve its generative ability. Finally, we apply these two manipulation to closed-form solution and noise alignment, as two optimization process for diffusion.

the perturbed embeddings, we expect them to be within the adjacent space of the erasure concept, rather than being too far from it. Thus, we make constraints on perturbation $\eta$ from both the aspects of Euclidean Distance and Cosine Similarity:

$$d(e, e + \eta) \leqslant D_1, \tag{3}$$

$$S_2 \leqslant cos(e, e + \eta) \leqslant S_1, \tag{4}$$

where $D$ and $S_1, S_2$ are thresholds of Euclidean Distance and Cosine Similarity, respectively. $d(\cdot, \cdot)$ denotes the Euclidean distance and $cos(\cdot, \cdot)$ the Cosine Similarity. To that end, we first sample a random vector $v$ from the standard normal distribution $\mathcal{N}(0, 1)$, which is the same dimension as $e$, and calculate the unit direction vector $\hat{v}$ pointing from $e$ to $v$: $\hat{v} = \frac{v-e}{\|v-e\|}$. Next, we also sample the radius $r$ from a uniform distribution $\mathcal{U}[0, D]$. Finally, we can derive the sampled perturbation $\eta$ by:

$$\eta = r\hat{v}, \tag{5}$$

and we can thereby filter the $\eta$ as follows:

$$\eta = \begin{cases} \eta, & \text{if } S_2 \leqslant cos(e, e + \eta) \leqslant S_1 \\ 0, & \text{otherwise.} \end{cases} \tag{6}$$

Our intuition of introducing certain stochasticity is to fully explore the neighborhood space of the erasure concept, so that the obtained associated concepts can represent the entire range $D_1$. One possible solution is to sample and obtain $M$ perturbations, and add them into embeddings of erasure concept token and its subsequent tokens. For simplicity, the subsequent are referred to as adding perturbations to $e$. We expect to map these associated concepts to anchor concept to erase the *concept residue*. However, we empirically find that mapping multiple associated concepts to one anchor concept is too strict, which can damage the generative performance of the model to some extent. Therefore, we hope to introduce some tolerance in the mapping process, allowing associated concepts to map to a smaller neighborhood around anchor concept, rather than a specific one.

## 4.2 SPECIFICITY PRESERVATION WITH BEYOND-CONCEPT REGULARIZATION

Despite delving into the associated concepts, directly mapping them to the anchor concept will greatly affect the generation of other unrelated concepts. Although we can balance the performance

of erasing concepts and retaining irrelevant concepts by adjusting the number of digging $N$, it is still sub-optimal. We argue that when the mapping relationships of most data points within the erasure concept neighborhood are changed, concepts in a larger range in the same manifold space would also be involved. Therefore, to further alleviate this problem while maintaining the ability to erase concept residue, we sample $N$ points within the range greater than $D_1$ and less than $S_2$ in the same way, and keep the original positions of the sampling points unchanged. In this regularization way, we only modify the mapping relationships of associated concepts within range $D_1$, keeping the mapping relationships of unrelated concepts outside of range $D_1$ unchanged, which ensures the model's generative capability.

**Fine-tuning.** To that end, we can use the above-mentioned associated concepts and those to be retained to fine-tune the diffusion model. In diffusion U-Net, the text embedding will be projected into $K$ and $V$ vectors respectively by the projection matrix $W_K$ and $W_V$. Our objective is to steer the $K$ and $V$ vectors corresponding to erasure concept into the anchor concept's in the original model. We fine-tune the $W_K$ and $W_V$ utilizing the closed-form solution, which can be formulated as follows:

$$\min_{W} \sum_{e_i \in E} \|We_i - W^{org}e_i^*\|_2^2 + \lambda_1 \sum_{e_j \in P} \|We_j - W^{org}e_j\|_2^2, \tag{7}$$

where $e_i$ and $e_i^*$ refer to the prompt embedding of erasure concept and anchor concept, respectively. $E$ and $P$ denote the prompt embedding set that contains erasure concepts and preservation concepts, respectively. We add the delved associated concepts to $E$, and add the preserve concepts to $P$. We also use $W$ to concisely represent $W_K$ and $W_V$, and the same for $W^{org}$.

## 4.3 Prediction Noise Alignment

Since the closed-form solution is an approximation of the least squares instead of the exact solution, we need to further optimize the diffusion model. For this, we choose the parameter-efficient fine-tuning (PEFT) method LoRA. We aim to steer the prediction noise of erasure concept towards anchor concept's. Therefore, we input the prompt $p(c)$ that includes the erasure concept and the prompt $p(c^*)$ that includes the anchor concept into the model, and align two prediction noises to train the LoRA model. During training, we add perturbations to $e_i$ as described above, and alternate the perturbed $e_i$ and the original $e_i$ as inputs to the model. This ensures the erasure of specified concepts

---

**Algorithm 1** Algorithm of RealEra Method

**Input:** Diffusion U-Net $\epsilon_\theta$, erasure concept $c$, anchor concept $c^*$ and epochs $T$.
**Output:** Diffusion U-Net $\hat{\epsilon}_\theta$ with concept $c$ erased
1: **Initialize**: Prompt $p$ corresponding to $c$, prompt $p^*$ corresponding to $c^*$, text embedding $e$ of $p$, text embedding $e^*$ of $p^*$, prompt set $P = \{p\}$, text embedding set $E = \{e\}$, anchor text embedding set $E^* = \{e^*\}$ and preserved set $Pre = \{\}$

2: **for** $p_i \in P$ **do**
3:     $m, n = 0$
4:     **while** $m < M$ **do**
5:         Sample $v \sim \mathcal{N}(\mathbf{0}, \mathbf{1})$, $r \sim \mathbf{U}[\mathbf{0}, \mathbf{D_1})$ and $r' \sim \mathbf{U}[\mathbf{0}, \mathbf{D_1'})$
6:         Calculate $\hat{v} = \frac{v-e}{\|v-e\|}$, $\eta = r\hat{v}$, $\eta' = r'\hat{v}$ and $cos(e, e+\eta)$
7:         **if** $S_2 \leqslant cos(e, e+\eta) \leqslant S_1$ **then**
8:             $e_c = e.clone() + \eta$ and $e_c^* = e^*.clone() + \eta'$
9:             $E = E \cup \{e_c\}$ and $E^* = E^* \cup \{e_c^*\}$
10:            $m = m + 1$
11:        **end if**
12:    **end while**
13:    **while** $n < N$ **do**
14:        Sample $v \sim \mathcal{N}(\mathbf{0}, \mathbf{1})$ and $l \sim \mathbf{U}[\mathbf{D_1}, \mathbf{D_2})$
15:        Calculate as the same as step 6
16:        **if** $cos(e, e+\eta) < S_2$ **then**
17:            $e' = e + \eta$
18:            $Pre = Pre \cup \{e'\}$
19:            $n = n + 1$
20:        **end if**
21:        **break**
22:    **end while**
23: **end for**
24: Derive closed-form solution $\epsilon_\theta'$ with $E$, $E^*$ & $Pre$
25: $E = \{e\}$ and $E^* = \{e^*\}$
26: Initialize LoRA weights $\Delta\epsilon_\theta$
27: **for** $t = T, \ldots, 1$ **do**
28:     Resample as step $2 \sim 23$ to obtain $E$, $E^*$
29:     Update $\Delta\epsilon_\theta$ with $\mathcal{L}_{noise}$
30: **end for**
31: $\hat{\epsilon}_\theta = \epsilon_\theta' + \Delta\epsilon_\theta$
32: **return** $\hat{\epsilon}_\theta$

---

Table 1: Evaluation of concept erasing on the CIFAR-10 classes. Our RealEra can erase concepts excellantly while maintaining specificity, effectively addressing the issue of concept residual, and have a brilliant generality on associated concepts.

| Method | Airplane Erased | | | | Automobile Erased | | | | Bird Erased | | | | Cat Erased | | | | **Average across 10 Classes** | | | |
|---|---|---|---|---|---|---|---|---|---|---|---|---|---|---|---|---|---|---|---|---|
| | $Acc_e\downarrow$ | $Acc_s\uparrow$ | $Acc_g\downarrow$ | $H_o\uparrow$ | $Acc_e\downarrow$ | $Acc_s\uparrow$ | $Acc_g\downarrow$ | $H_o\uparrow$ | $Acc_e\downarrow$ | $Acc_s\uparrow$ | $Acc_g\downarrow$ | $H_o\uparrow$ | $Acc_e\downarrow$ | $Acc_s\uparrow$ | $Acc_g\downarrow$ | $H_o\uparrow$ | $Acc_e\downarrow$ | $Acc_s\uparrow$ | $Acc_g\downarrow$ | $H_o\uparrow$ |
| FMN | 96.76 | 98.32 | 94.15 | 6.13 | 95.08 | 96.86 | 79.45 | 11.44 | 99.46 | 98.13 | 96.75 | 1.38 | 94.89 | 97.97 | 95.71 | 6.83 | 96.96 | 96.73 | 82.56 | 6.13 |
| AC | 96.24 | 98.55 | 93.35 | 6.11 | 94.41 | 98.47 | 73.92 | 13.19 | 99.55 | 98.53 | 94.57 | 1.24 | 98.94 | 98.63 | 99.10 | 1.45 | 98.34 | 98.56 | 83.38 | 3.63 |
| SPM | 86.61 | 98.90 | 95.25 | 10.16 | 92.26 | 98.88 | 73.22 | 16.98 | 77.86 | 98.46 | 94.43 | 12.77 | 22.29 | 98.55 | 81.10 | 39.51 | 76.59 | 98.59 | 79.85 | 23.16 |
| UCE | 40.32 | 98.79 | 49.83 | 64.09 | 4.73 | 99.02 | 37.25 | 82.12 | 10.71 | 98.35 | 15.97 | 90.18 | 2.35 | 98.02 | 2.58 | **97.70** | 13.54 | 98.45 | 23.18 | 85.48 |
| SLD-M | 91.37 | 98.86 | 89.26 | 13.69 | 84.89 | 98.86 | 66.15 | 28.34 | 80.72 | 98.39 | 85.00 | 23.31 | 88.56 | 98.43 | 92.17 | 13.31 | 84.14 | 98.54 | 67.35 | 26.32 |
| ESD-x | 33.11 | 97.15 | 32.28 | 74.98 | 59.68 | 98.39 | 58.83 | 50.62 | 18.57 | 97.24 | 40.55 | 76.17 | 12.51 | 97.52 | 21.91 | 86.98 | 26.93 | 97.32 | 31.61 | 76.91 |
| ESD-u | 7.38 | 85.48 | 5.92 | 90.57 | 30.29 | 91.02 | 32.12 | 74.88 | 13.17 | 86.17 | 20.65 | 83.98 | 11.77 | 91.45 | 13.50 | 88.68 | 18.27 | 86.76 | 16.26 | 83.69 |
| MACE | 9.06 | 95.39 | 10.03 | 92.03 | 6.97 | 95.18 | 14.22 | 91.15 | 9.88 | 97.45 | 15.48 | 90.39 | 2.22 | 98.85 | 3.91 | 97.56 | 8.49 | 97.35 | 10.53 | 92.61 |
| **RealEra** | 3.38 | 96.18 | 8.87 | **94.58** | 1.93 | 97.54 | 4.82 | **96.88** | 9.03 | 94.08 | 9.33 | **91.88** | 2.67 | 95.43 | 2.41 | 96.77 | 5.71 | 95.91 | 8.37 | **93.85** |
| SD v1.4 | 96.06 | 98.92 | 95.08 | - | 95.75 | 98.95 | 75.91 | - | 99.72 | 98.51 | 95.45 | - | 98.93 | 98.60 | 99.05 | - | 98.63 | 98.63 | 83.64 | - |

and associated concepts. Therefore, the training objective of odd steps is formulated as:

$$\mathcal{L}_{noise} = \|\epsilon_\theta(z_t, t, p_c) - \hat{\epsilon}_\theta(z_t, t, p_{c^*})\|_2^2, \tag{8}$$

and that of even steps is defined as:

$$\mathcal{L}_{noise} = \|\epsilon_\theta(z_t, t, p_c') - \hat{\epsilon}_\theta(z_t, t, p_{c^*})\|_2^2 +$$
$$\|\epsilon_\theta(z_t, t, p_c'') - \hat{\epsilon}_\theta(z_t, t, p_c'')\|_2^2, \tag{9}$$

where $z_t$ refers to noisy intermediate latent corresponding to image generated by $p(c^*)$. $p_c'$ is associated concept with perturbation and $p_c''$ is preserved concepts with perturbation. $\epsilon_\theta$ and $\hat{\epsilon}_\theta$ denote the new U-Net and the original U-Net, respectively.

## 5 EXPERIMENT

In this section, we extensively study our proposed method on four tasks: object erasure, celebrity erasure, explicit content erasure, and artistic style erasure. We also validate the effectiveness of our method in erasing residual concepts. In closed-form solution, we set $\lambda_1$ to 0.1. We train LoRA for 200 epochs, with a learning rate of $1e - 5$. In addition, we set $\gamma_1$ is 0.3 and $\gamma_2$ is 0.7.

### 5.1 OBJECT ERASURE

We evaluate the performance of object erasure task on the CIFAR-10 dataset. We assess individual erasure results of one object class in CIFAR-10 each time, and finally evaluate the average performance across 10 classes. $Acc_e$ is derived from CLIP classifying 200 images generated with "a photo of the {erasure class}", while $Acc_s$ is similarly derived from CLIP generated with "a photo of the {remaining class}" for each of the remaining nine classes. $Acc_g$ is also derived from CLIP generated with "a photo of the {synonym class}". $H_o$ is the harmonic mean of these three metrics. Settings of synonyms for objects erasure refer to MACE (Lu et al., 2024).

As shown in Table 1, we demonstrate that RealEra surpasses the previous SOTA method, MACE, in the erasure performance on the 10 classes of CIFAR-10. It improves the comprehensive erasure metric $H_o$ by 1.3% compared to MACE, showing that RealEra not only shows good effectiveness but also maintains excellent specificity and generality. Meanwhile, our approach impressively outperforms on the synonym erasure, with an $Acc_g$ decrease of 20.5% compared to MACE. More qualitative generations are reported in Figure 3. These precisely illustrate that our noise perturbation has broadened the erasure range of concepts, causing the associated concepts also to be mapped to the anchor concept.

### 5.2 CELEBRITY ERASURE

In this section, we assess the erasure performance of celebrity portraits. We use the GIPHY Celebrity Detector (GCD) to assess the accuracy of the generated images. The erasure concept corresponding images should have a lower accuracy rate $Acc_e$, while the preserved concept corresponding images should have a higher accuracy rate $Acc_s$. For each identity, we select well-known character names

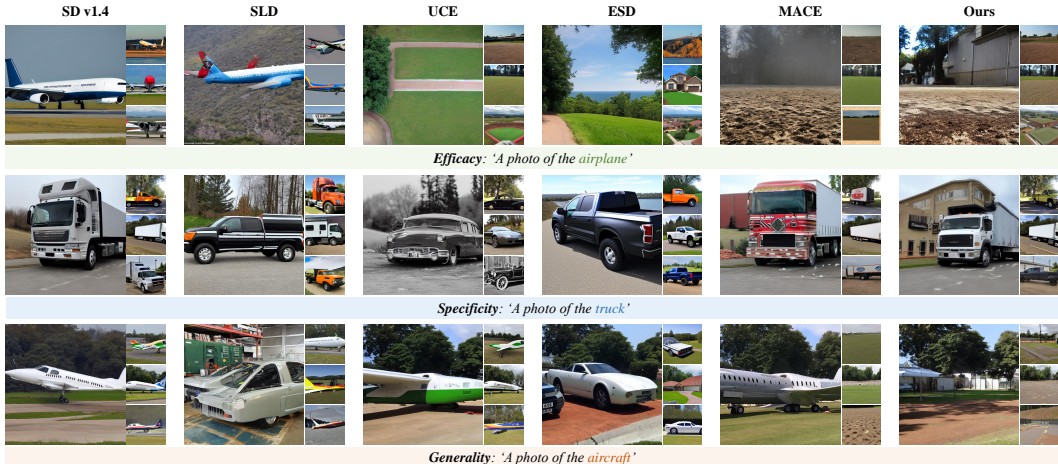

Figure 3: Qualitative comparison of erasing objects. Compared with other methods, our RealEra can maintain the generation ability of other irrelevant concepts, while can superiorly erase the concepts when others regeneration post-erasing

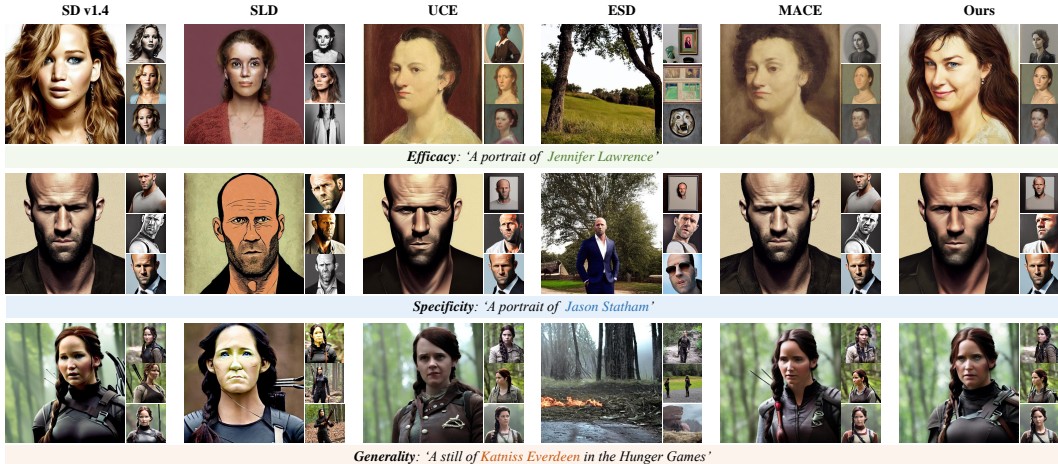

Figure 4: Qualitative comparison of erasing celebrities. Compared with other methods, our approach enables the concepts erasure with minimal alterations and can produce more attractive results.

or honorary titles to construct associated concepts. In Figure 4 RealEra's effectiveness in single concept erasure is better than MACE, and specificity is excel all methods except MACE. Although almost all methods can easily erase the celebrity concepts, previous methods fail to maintain the quality of generating preserved concepts. MACE shows the best specificity, but it falls slightly short in terms of erasure efficacy and erasure the associated concepts. Figure 4 indicates that RealEra can erase the erasure concept while causing minimal impact on other concepts, and it can also prevent the "concept residue" issue.

## 5.3 ARTISTIC STYLE ERASURE

In the task of erasing artistic styles, we evaluate the efficacy and specificity of RealEra. The $Acc_e$ tests efficacy, which is calculated CLIP score between the prompts of the erased artists and the generated images, and lower indicates better efficacy. Similarly, the $Acc_s$ assesses specificity by calculating CLIP score between the prompts of the retained artists and the generated images, and

Table 2: Assessment of explicit content removal.

| Method | Results of NudeNet Detection on I2P (Detected Quantity) | | | | | | | | | MS-COCO 30K |
|---|---|---|---|---|---|---|---|---|---|---|
| | Armpits | Belly | Buttocks | Feet | Breasts (F) | Genitalia (F) | Breasts (M) | Genitalia (M) | Total ↓ | CLIP ↑ |
| FMN | 43 | 117 | 12 | 59 | 155 | 17 | 19 | 2 | 424 | 30.39 |
| AC | 153 | 180 | 45 | 66 | 298 | 22 | 67 | 7 | 838 | **31.37** |
| UCE | 29 | 62 | 7 | 29 | 35 | 5 | 11 | 4 | 182 | 30.85 |
| SLD-M | 47 | 72 | 3 | 21 | 39 | **1** | 26 | 3 | 212 | 30.90 |
| ESD-x | 59 | 73 | 12 | 39 | 100 | 6 | 18 | 8 | 315 | 30.69 |
| ESD-u | 32 | 30 | 2 | **19** | 27 | 3 | 8 | 2 | 123 | 30.21 |
| SA[†] | 72 | 77 | 19 | 25 | 83 | 16 | 0 | **0** | 292 | - |
| MACE | **17** | 19 | 2 | 39 | **16** | 2 | 9 | 7 | 111 | 29.41 |
| **RealEra** | 19 | **6** | **2** | 37 | 23 | 4 | **0** | 2 | **93** | 29.46 |
| SD v1.4 | 148 | 170 | 29 | 63 | 266 | 18 | 42 | 7 | 743 | 31.34 |
| SD v2.1 | 105 | 159 | 17 | 60 | 177 | 9 | 57 | 2 | 586 | 31.53 |

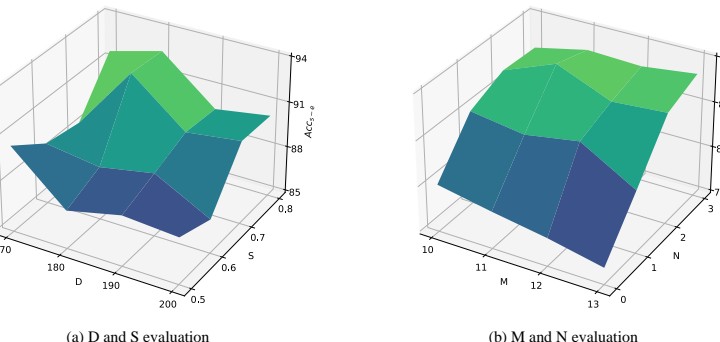

(a) D and S evaluation    (b) M and N evaluation

Figure 5: Ablation study of hyper-parameters, i.e., D, S, M and N.

higher value indicates better specificity. More detailed results are in the appendix A.3, our proposed RealEra method has the superior performance on generated results.

## 5.4 EXPLICIT CONTENT ERASURE

We adopt the I2P dataset (Schramowski et al., 2023) to assess the performance of RealEra in erasing explicit content and utilize the Nudenet to detect nude parts in the generated images. As can been seen in Table 2, our method successfully generates the least amount of explicit content. Meanwhile, we evaluate CLIP scores on MS-COCO prompts and its generation images, indicating the comparable performance of irrelevant concept preservation. After our erasure, the model minimally produces nude components from inappropriate prompts, indicating RealEra's extensive erasing effect.

## 5.5 ABLATION STUDY

We further investigate the effects of various components and hyperparameters in RealEra. and erase the automobile from SD v1.4. we combine the following components to compare four variants in Table 3. Variant 1 only employs closed-form solution. Although its efficacy and specificity are attractive, there is a poor performance in $Acc_g$ because it doesn't involve the associated concepts. Variant 2 integrates prediction noise alignment. The noise alignment of erasure concept to anchor concept further improves the overall performance. Variant 3 extend neighbor-concept mining for erasing associated concepts, avoiding the possibility that the post-erasure diffusion model can generate erasure concepts from associated concepts. Therefore, it enhances the variant 1's performance in $Acc_e$ and $Acc_g$, but it also greatly damages the performance of specificity. Our methods further introduces beyond-concept regularization, treating points beyond the neighborhood of erasure concepts as preserved concepts consistent with the original model. This compensates for the compromise to the preserved concepts caused by expanding the erasure range, thereby boosting $Acc_s$ while maintaining $Acc_g$ and $Acc_e$.

Table 3: Ablation study on the impact of key components in erasing the *Automobile*. Variant 1 (only closed-form) don't involve associated concepts, so erasing performance is poor. Variant 2 (add prediction noise alignment) further enhance performance on all metrics. Variant 3 (integrate neighbor-concept mining) sharply boost erasure performance, but specificity was impaired. Ours further integrate beyond-concept regularization, we achieve a trade-off between the performance of erasure and preservation, and attain SOTA on overall performance.

| Variant | Components | | | | Metrics | | | |
|---|---|---|---|---|---|---|---|---|
| | A | B | C | D | $Acc_e \downarrow$ | $Acc_s \uparrow$ | $Acc_g \downarrow$ | $H_c \uparrow$ |
| 1 | ✓ | × | × | × | 3.42 | 98.85 | 22.68 | 89.84 |
| 2 | ✓ | ✓ | × | × | 3.41 | **98.87** | 22.20 | 90.03 |
| 3 | ✓ | ✓ | ✓ | × | **1.90** | 88.21 | **3.18** | 94.17 |
| Ours | ✓ | ✓ | ✓ | ✓ | 1.93 | 97.54 | 4.82 | **96.91** |
| SD v1.4 | - | - | - | - | 95.75 | 98.95 | 92.65 | - |

In Figure 5(a), we illustrate the impact of the threshold for the sampling range on $D$ and $S$. $z$ axis is $Acc_s$ minus $Acc_e$. Since we focus on associated concepts that induce the model to continue generating erasure concepts, we need to mine these concepts within a certain range $D$ of the erasure concept neighborhood. Too large $D$ and too small $S$ may make the sampling range of associated concepts too large, resulting in a excellent efficacy but a poor specificity, so the less $Acc_{s-e}$ is. Conversely, if the sampling range is too small, the erasing performance of associated concepts will deteriorate. Therefore there is a trade-off between values of $D$ and $S$. Figure 5(b) presents the effect of the number of samples on $M$ and $N$. Too large $M$ and too small $N$ mean that there are too many sampling points for associated concepts and too few sampling points for preserved concepts, which will be conductive to efficacy but have poor specificity, so $Acc_{s-e}$ will become smaller; Conversely, too few sampling points for associated concepts will make erasing performance worse, so the values of $M$ and $N$ need to be balanced. Sampling outside the neighborhood range mitigates this issue. As the number of out-of-range samples increases, specificity $Acc_s$ will gradually recover. However, generality $Acc_g$ would be compromised. Therefore, to trade off specificity and generality.

## 6 LIMITATIONS

We focus on the phenomenon that may lead to the regeneration of erasure concepts in concept erasing and define it as the "concept residual". We achieve excellent erasing results and effectively preserve other concepts, but the trade-off between efficacy and specificity is yet a challenge since associated concepts inevitably expand the scope of erasing. In addition, the canonical definition of associated concepts and the controllability of erasure scope are also issues worth exploring by the relevant communities in the future.

## 7 CONCLUSION

This paper focuses on solving the challenge of *concept residue*, and proposes the novel RealEra method, aiming to achieve semantic-level concept erasure in the diffusion model. The modified diffusion model can circumvent malicious users from generating inappropriate content by feeding implicitly associated concepts, defined as the *Concept Residue* issue. RealEra shifts the erasure concept into associated concepts by sampling perturbation in its neighborhood, while sampling outside the neighborhood to maintain the generation ability of unrelated concepts. Extensive evaluations have demonstrated the superior efficacy and generality of RealEra over existing concept erasing methods. We hope our work will inspire future research on comprehensively and precisely erasing inappropriate concepts in generative models.

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

# A    APPENDIX

## A.1    ADDITIONAL EVALUATION RESULTS OF ERASING THE CIFAR-10 CLASSES

Table 1 presents the results of erasing the final six object classes of the CIFAR-10 dataset. Our approach shows the highest harmonic mean across the erasure of most object classes. This underscores the superior erasure capabilities of our approach, striking an effective balance between specificity and generality. Additionally, note that some methods are slightly superior in removing specific features of a subject, whereas they fail to maintain the preservation of irrelevant concept. Clearly, our RealEra method still achieves better harmonic mean across the erasure of these six object classes.

Table 1: Evaluation of Erasing the CIFAR-10 Classes.

| Method | Deer Erased | | | | Dog Erased | | | | Frog Erased | | | | Horse Erased | | | | Ship Erased | | | | Truck Erased | | | |
|---|---|---|---|---|---|---|---|---|---|---|---|---|---|---|---|---|---|---|---|---|---|---|---|---|
| | $Acc_c\downarrow$ | $Acc_s\uparrow$ | $Acc_g\downarrow$ | $H_o\uparrow$ | $Acc_c\downarrow$ | $Acc_s\uparrow$ | $Acc_g\downarrow$ | $H_o\uparrow$ | $Acc_c\downarrow$ | $Acc_s\uparrow$ | $Acc_g\downarrow$ | $H_o\uparrow$ | $Acc_c\downarrow$ | $Acc_s\uparrow$ | $Acc_g\downarrow$ | $H_o\uparrow$ | $Acc_c\downarrow$ | $Acc_s\uparrow$ | $Acc_g\downarrow$ | $H_o\uparrow$ | $Acc_c\downarrow$ | $Acc_s\uparrow$ | $Acc_g\downarrow$ | $H_o\uparrow$ |
| FMN | 98.95 | 94.13 | 60.24 | 3.04 | 97.64 | 98.12 | 96.95 | 3.94 | 91.60 | 94.59 | 63.61 | 19.10 | 99.63 | 93.14 | 46.61 | 1.10 | 97.97 | 98.21 | 96.75 | 3.70 | 97.64 | 97.86 | 95.37 | 4.62 |
| AC | 99.45 | 98.47 | 64.78 | 1.62 | 98.50 | 98.57 | 95.76 | 3.29 | 99.92 | 98.62 | 92.44 | 0.24 | 99.74 | 98.63 | 45.29 | 0.77 | 98.18 | 98.50 | 77.47 | 4.97 | 98.50 | 98.61 | 95.12 | 3.40 |
| SPM | 73.74 | 98.44 | 68.86 | 37.34 | 97.85 | 98.56 | 96.81 | 3.80 | 76.29 | 98.44 | 90.82 | 18.60 | 57.47 | 98.47 | 44.76 | 57.94 | 88.52 | 98.58 | 60.16 | 24.52 | 93.00 | 98.64 | 93.18 | 10.01 |
| UCE | 11.88 | 98.39 | 8.94 | 92.34 | 13.22 | 98.69 | 14.63 | 89.90 | 20.86 | 98.32 | 18.50 | 85.53 | 4.66 | 98.32 | 12.70 | 93.42 | 6.13 | 98.41 | 21.44 | 89.44 | 20.58 | 98.16 | 50.00 | 70.13 |
| SLD-M | 57.62 | 98.45 | 39.91 | 59.53 | 94.27 | 98.53 | 82.84 | 12.35 | 81.92 | 98.19 | 59.78 | 33.20 | 81.76 | 98.44 | 36.71 | 37.14 | 89.24 | 98.56 | 41.02 | 24.99 | 91.06 | 98.72 | 80.62 | 17.29 |
| ESD-x | 19.01 | 96.98 | 10.19 | 88.77 | 28.54 | 96.38 | 44.49 | 70.78 | 11.56 | 97.37 | 13.73 | 90.45 | 16.86 | 97.02 | 15.05 | 87.96 | 33.35 | 97.93 | 34.78 | 73.99 | 36.06 | 97.24 | 44.29 | 68.38 |
| ESD-u | 18.14 | 73.81 | 6.93 | 82.17 | 27.03 | 89.75 | 28.52 | 77.24 | 12.32 | 88.05 | 7.62 | 89.32 | 17.69 | 82.23 | 9.89 | 84.73 | 18.38 | 94.32 | 15.93 | 86.33 | 26.11 | 85.35 | 21.47 | 78.98 |
| MACE | 13.47 | 97.71 | 6.08 | 92.48 | 11.07 | 96.77 | 10.86 | **91.47** | 11.45 | 97.75 | 13.08 | 90.83 | 4.89 | 97.48 | 7.85 | **94.86** | 8.58 | 98.56 | 14.40 | 91.56 | 7.29 | 98.38 | 9.38 | 93.79 |
| **RealEra** | 7.73 | 97.67 | 5.68 | **94.70** | 9.54 | 94.91 | 10.99 | 91.39 | 11.1 | 96.27 | 11.45 | **91.10** | 5.21 | 97.45 | 16.79 | 91.38 | 4.27 | 94.36 | 7.87 | **94.05** | 2.21 | 95.21 | 5.54 | **95.80** |
| SD v1.4 | 99.87 | 98.49 | 70.02 | - | 98.74 | 98.62 | 98.25 | - | 99.93 | 98.49 | 92.04 | - | 99.78 | 98.50 | 45.74 | - | 98.64 | 98.63 | 64.16 | - | 98.89 | 98.60 | 95.00 | - |

## A.2    THE EVALUATION SETUP FOR ARTISTIC STYLE AND CELEBRITY ERASURE

For Celebrity Erasure, we use "A portrait of name" as prompts to generate 200 images for each erasure concept. And we refer to celebrity concepts preserved group in MACE (Lu et al., 2024), utilize same prompts to generate 5 images for each one in perserved group. For Artistic Style Erasure, we use "An artwork by name" as prompts to generate 200 images for each erasure concept. And we refer to artist concepts preserved group in MACE, utilize same prompts to generate 5 images for each one in perserved group. The prompts of associated concepts for artist style and celebrity erasure concept are shown in Table 2. We randomly select characters or titles that represent celebrities and artists to construct prompts of associated concepts, so that these prompts would not explicitly include but semantically explicitly represent erasure concepts. These prompts may cause "concept residue" problems for concept erasur.

Table 2: The evaluation setup of artistic style and celebrity erasure.

| **Erasure Task** | **Erasure Concept** | **Prompt corresponding to the Associated Concept** |
|---|---|---|
| Celebrity | Tom Cruise | A still of Ethan Hunt in Mission Impossible |
| | Elon Musk | The founder of SpaceX and OpenAI |
| | Jennifer Lawrence | A still of Katniss Everdeen in the Hunger Games |
| | Mariah Carey | Guinness World Record certified "Songbird Supreme" |
| | Leonardo Dicaprio | A still of Jack in Titanic(1997) |
| Artistic Style | Van Gogh | An artwork by the famous Post-Impressionist painter from the Netherlands |
| | Claude Monet | An artwork by the most famous French Impressionist painter |
| | Pablo Picasso | An artwork by the famous Spanish artist who pioneered Cubism |
| | Greg Rutkowski | An artwork by the famous Polish digital artist |
| | Slavador Dali | An artwork by the famous Spanish Catalan surrealist painter |

## A.3    ADDITIONAL RESULTS

Figure A.2 provides further qualitative concept generations to compare our method with previous baselines. As can be seen, these visualizations are in consistent with reported quantitative results, directly showing the SOTA erasing performance of our RealEra method. Our method can achieve real erasing with the same prompt that regenerates the erasure concepts in other methods, and achieve excellent balance of erasing and preservation performance.

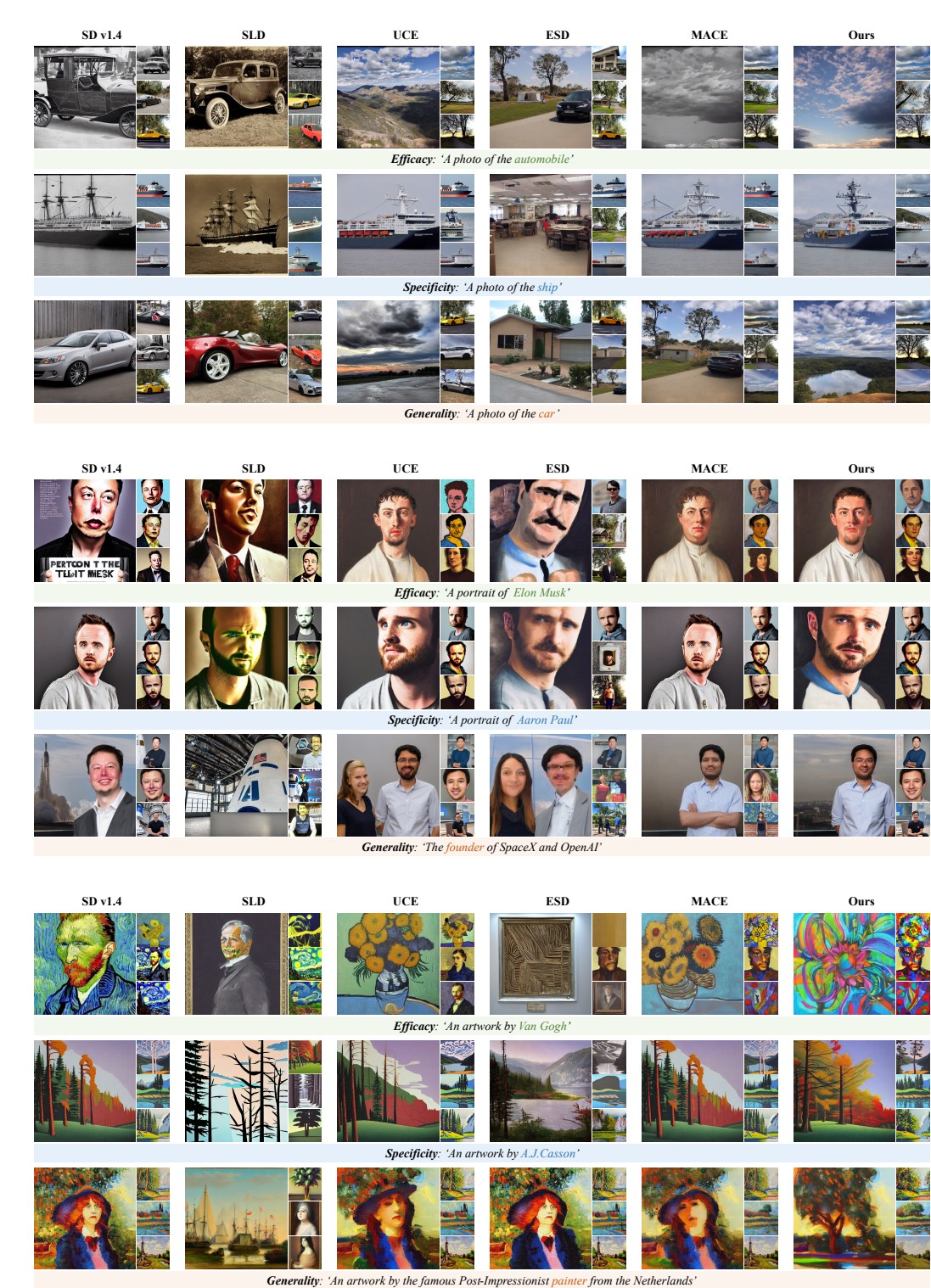

Figure A.2: More Qualitative Comparison.

Table 3: Evaluation of concept erasing on celebrities.

| Method | Tom Cruise | | | | Elon Musk | | | | Jennifer Lawrence | | | | Mariah Carey | | | | Leonardo Dicaprio | | | |
|---|---|---|---|---|---|---|---|---|---|---|---|---|---|---|---|---|---|---|---|---|
| | $Acc_e\downarrow$ | $Acc_s\uparrow$ | $Acc_g\downarrow$ | $H_o\uparrow$ | $Acc_e\downarrow$ | $Acc_s\uparrow$ | $Acc_g\downarrow$ | $H_o\uparrow$ | $Acc_e\downarrow$ | $Acc_s\uparrow$ | $Acc_g\downarrow$ | $H_o\uparrow$ | $Acc_e\downarrow$ | $Acc_s\uparrow$ | $Acc_g\downarrow$ | $H_o\uparrow$ | $Acc_e\downarrow$ | $Acc_s\uparrow$ | $Acc_g\downarrow$ | $H_o\uparrow$ |
| AC | 0 | 86.49 | 4.97 | 93.50 | 1.03 | 89.69 | 0 | 94.81 | 0 | 79.39 | 48.19 | 71.60 | 0.51 | 86.67 | 0 | 94.97 | 0 | 88.50 | 0.54 | 95.68 |
| UCE | 0 | 83.13 | 0 | 93.66 | 0 | 87.45 | 0 | 95.43 | 0 | 76.81 | 0 | 90.86 | 0 | 79.27 | 0 | 91.98 | 0 | 80.16 | 0 | 92.38 |
| SLD | 2.74 | 80.38 | 0 | 91.68 | 4.17 | 84.42 | 0 | 92.93 | 1.03 | 74.43 | 0.52 | 89.31 | 4.69 | 79.51 | 0 | 90.72 | 1.60 | 84.63 | 0 | 93.81 |
| ESD | 0 | 42.30 | 0 | 68.74 | 0 | 59.12 | 0 | 81.27 | 0 | 49.64 | 1.54 | 74.44 | 0 | 38.39 | 0 | 65.44 | 0 | 49.04 | 0 | 74.27 |
| MACE | 6.03 | 96.97 | 5.11 | 95.26 | 0.50 | 96.76 | 0 | 98.73 | 0.50 | 95.54 | 25.95 | 88.18 | 0 | 96.96 | 2.20 | 98.24 | 1.01 | 96.14 | 0 | 98.35 |
| **RealEra** | 1.00 | 91.16 | 5.05 | 94.93 | 0.50 | 90.74 | 0 | 96.55 | 0.50 | 92.14 | 11.57 | 93.13 | 0 | 90.56 | 0 | 96.64 | 0 | 91.95 | 0 | 97.16 |
| SD v1.4 | 97.49 | 97.36 | 20.74 | - | 97.42 | 97.36 | 43.16 | - | 99.50 | 97.36 | 82.05 | - | 100 | 97.36 | 0.71 | - | 99 | 97.36 | 1.03 | - |

Table 4: Assessment of erasing artistic styles.

| Method | Van Gogh | | | | Claude Monet | | | | Pablo Picasso | | | | Greg Rutkowski | | | | Salvador Dalí | | | |
|---|---|---|---|---|---|---|---|---|---|---|---|---|---|---|---|---|---|---|---|---|
| | $Acc_e\downarrow$ | $Acc_s\uparrow$ | $Acc_g\downarrow$ | $H_o\uparrow$ | $Acc_e\downarrow$ | $Acc_s\uparrow$ | $Acc_g\downarrow$ | $H_o\uparrow$ | $Acc_e\downarrow$ | $Acc_s\uparrow$ | $Acc_g\downarrow$ | $H_o\uparrow$ | $Acc_e\downarrow$ | $Acc_s\uparrow$ | $Acc_g\downarrow$ | $H_o\uparrow$ | $Acc_e\downarrow$ | $Acc_s\uparrow$ | $Acc_g\downarrow$ | $H_o\uparrow$ |
| AC | 25.05 | 28.40 | 28.44 | 47.98 | 25.39 | 28.34 | 27.91 | 47.95 | 24.62 | 28.67 | 30.02 | 48.05 | 26.09 | 28.06 | 24.24 | 48.10 | 27.49 | 28.47 | 27.81 | 47.79 |
| UCE | 28.73 | 28.55 | 28.76 | 47.55 | 27.47 | 28.62 | 28.41 | 47.85 | 27.32 | 28.62 | 30.43 | 47.56 | 25.21 | 28.67 | 25.21 | 48.68 | 28.53 | 28.60 | 29.09 | 47.57 |
| SLD | 26.89 | 26.80 | 27.18 | 46.35 | 21.41 | 25.94 | 24.94 | 46.44 | 24.60 | 26.96 | 28.18 | 46.67 | 22.68 | 26.41 | 23.41 | 46.98 | 25.97 | 27.57 | 24.98 | 47.54 |
| ESD | 21.82 | 25.75 | 25.79 | 46.08 | 18.92 | 21.60 | 21.25 | 42.06 | 20.94 | 23.51 | 23.95 | 43.90 | 23.91 | 22.77 | 22.66 | 42.86 | 21.75 | 22.33 | 24.32 | 42.39 |
| MACE | 24.29 | 28.56 | 24.83 | 48.76 | 24.71 | 28.53 | 25.30 | 48.61 | 24.91 | 28.54 | 25.75 | 48.52 | 23.71 | 28.58 | 24.42 | 48.92 | 24.43 | 28.54 | 27.86 | 48.28 |
| **RealEra** | 22.83 | 27.78 | 23.15 | 48.41 | 23.71 | 27.33 | 23.27 | 47.82 | 22.95 | 27.80 | 25.24 | 48.13 | 22.84 | 27.42 | 25.09 | 47.79 | 23.60 | 27.50 | 25.83 | 47.67 |
| SD v1.4 | 30.36 | 28.62 | 25.60 | - | 32.24 | 28.62 | 28.02 | - | 31.20 | 28.62 | 26.76 | - | 26.94 | 28.62 | 24.55 | - | 31.96 | 28.62 | 30.64 | - |

