# OpenReview forum: "RealEra:Semantic-level Concept Erasure via Neighbor-Concept Mining"
_ICLR.cc/2025/Conference — ICLR 2025 Conference Withdrawn Submission_

### Official Review · Reviewer_hqqo · 2024-10-30

**Soundness:** 2
**Presentation:** 2
**Contribution:** 2
**Rating:** 3
**Confidence:** 4

**Summary:**

This work addresses the "concept residue issue," where existing erasing methods may still implicitly regenerate the target concept even after it has been erased. To tackle this, a method called "neighbor-concept mining" is proposed, which applies random noise perturbation to the erasing concepts, enabling the removal of concepts associated with the erased ones. This approach allows for the erasure of implicit concepts. Additionally, a preservation loss called "beyond-concept regularization" is devised to protect areas beyond the neighborhood of the erasing concept. Experiments demonstrate that this method can improve the generalizability of concept erasure.

**Strengths:**

1. The paper was well written and easy to follow.
2. The straightforward method shows effectiveness for improving the generality on erasing concepts

**Weaknesses:**

1. The task is not novel and lacks sufficient comparison to other baselines. Improving the generality for concept erasing has been studied in previous papers beyond just MACE. For example, Geom-Erasing [1] was proposed to enhance the generality of erasing implicit concepts, but this paper does not compare to such baselines. Please compare your method with additional methods, including Geom-Erasing [1] and Receler [4], and explain the primary differences between them.

2. The beyond-concept regularization is also not novel, as it was similarly proposed in SPM, which uses preservation loss for random embeddings with low similarity to erased concepts. However, this paper does not clarify the differences in preservation methods between it and SPM, nor does it provide comparative experiments on this aspect. Please provide the key distinctions between beyond-concept regularization and SPM's preservation loss, supported by comparative experiments and an explicit explanation.

3. The performance of the proposed method is not very convincing. In Figure 4, the qualitative results for erasing Jennifer Lawrence do not strongly suggest that Katniss Everdeen was removed in terms of generality. Particularly, in Tables 3 and 4 in Appendix A.3, MACE clearly outperforms the proposed method in terms of the H_o score. Overall, it seems that much specificity has been sacrificed to improve generality. This suggests that the proposed method may not enhance overall performance or provide a solution to the trade-off between generality and specificity.

4. In this study, Figure 5 shows how generality and specificity change depending on D and S. However, there is no explanation for the criteria used to select D and S or how they were compared to other baselines. As these values determine the trade-off between generality and specificity, they are closely related to the paper's argument, but this consideration is lacking. Please provide a detailed explanation of the criteria used to select D and S values and describe how these choices compare to parameter selections in other baselines. Additionally, explain how these parameters impact the trade-off between generality and specificity in relation to the paper's central argument.

5. The experimental setup is limited. Recently, robustness to attack prompts in concept erasing has become crucial for safe diffusion models [2,3]. This is intended to prevent the circumvention of erased concepts' regeneration, which is closely related to generality, yet the paper does not conduct experiments on this. Additionally, the paper only addresses erasing in a single concept erasure environment, raising doubts about whether performance would be maintained in a massive concept erasing application like MACE. For instance, in Tables 3 and 4 of Appendix A.3, there is a significant drop in specificity performance with just a single concept erasure for the proposed method. Concerns remain that specificity might be substantially compromised if 100 concepts are erased simultaneously, as with MACE.

[1] Zhili Liu, et al.,”Implicit Concept Removal of Diffusion Models”, ECCV, 2024
[2] Tsai, Yu-Lin, et al. "Ring-A-Bell! How Reliable are Concept Removal Methods For Diffusion Models?." ICLR. 2024.
[3]  Chao Gong, et al., Reliable and efficient concept erasure of text-to-image diffusion models, ECCV, 2024
[4] Huang, Chi-Pin, et al. "Receler: Reliable concept erasing of text-to-image diffusion models via lightweight erasers." ECCV, 2024.

**Questions:**

Please, see weakness.

---

### Official Review · Reviewer_PZma · 2024-11-01

**Soundness:** 1
**Presentation:** 2
**Contribution:** 1
**Rating:** 1
**Confidence:** 4

**Summary:**

This paper aims to remove harmful concepts from text-to-image generation models. For a given concept, the authors randomly select closely related neighboring concepts. To achieve this, they constrain the Euclidean distance and cosine similarity between the original concept and its neighbors. However, this removal process inadvertently erases other unrelated concepts. To preserve these unrelated concepts, the authors identify additional concepts that remain near the original concept but are slightly farther from the erased neighbors.

**Strengths:**

The papers utilize Euclidean distance and cosine similarity to constrain the neighboring search space, and the removal of neighboring concepts supports the overall elimination of the original concept. This method effectively reduces the search space.

**Weaknesses:**

1. Missed Related Works. Rather than using random generation for neighboring concepts, [A] applies adversarial training. These two methods are similar. The authors should include [A] as a baseline method.

2. Abnormal Experimental Results. Table 2 shows that AC performs worse than SD v1.4, whereas Table 1 in [A] indicates that AC significantly outperforms SD v1.4. Therefore, the authors should report more hyper-parameters and discuss why this study reaches a different conclusion compared to previous work. Besides, RealEra generates 93 nude images, exceeding the 66 generated by [A].

[A] Gong C, Chen K, Wei Z, et al. Reliable and efficient concept erasure of text-to-image diffusion models[J]. arXiv preprint arXiv:2407.12383, 2024.

**Questions:**

The primary concern lies in the experimental settings. The authors should identify where errors may have occurred in aligning the experimental conclusions.

---

### Official Review · Reviewer_6YLN · 2024-11-01

**Soundness:** 2
**Presentation:** 2
**Contribution:** 2
**Rating:** 5
**Confidence:** 4

**Summary:**

The paper introduces RealEra, a framework for semantic-level concept erasure in text-to-image models. It addresses the "concept residue" issue, where models inadvertently generate unwanted content linked to semantically associated inputs. RealEra uses neighbor-concept mining to identify and erase these related concepts while maintaining specificity through beyond-concept regularization. This ensures unrelated concepts retain their generative performance. The framework employs a closed-form solution and LoRA module for optimization. Experiments demonstrate that RealEra surpasses previous methods in erasing efficacy, specificity, and generality.

**Strengths:**

- Controlling the output of text-to-image diffusion models is an important research topic, which should be aligned with the values of the general public.
- The research problem studied in this manuscript is significant. In text-to-image models, concepts are described by words and sentences, while there are some "one-to-many" relationships, and different sentences and words can be used to describe the same concept. Erasing the target concept regardless of the descriptive prompts is a significant research challenge.

**Weaknesses:**

- There is no theoretical guarantee about whether related but non-target concepts would be affected by this proposed method. It is concerning that only limited empirical results support the specificity of the proposed method, and we will never know if there exists a prompt about irrelevant concepts that will be impaired by the proposed method.
- The proposed method adds perturbations to the target input embedding to find associated concepts in the adjacent semantic space, while the manuscript does not justify whether it can really find these associated concepts based on the metrics of Euclidean Distance and Cosine Similarity.
- See the questions below for more details.

**Questions:**

- I have questions about the motivation of concept erasing or machine unlearning in text-to-image diffusion models. As you said, NSFW safety checkers can be easily bypassed since the parameters and codes are publicly available. However, if the model user simply modifies or fine-tunes the safety checkers on other datasets, such bypass can be avoided due to the changed parameters. I wonder if the story of using erasure for undesired generation prevention is still convincing.
- How to precisely unlearn the target concept? You say the proposed method can steer the associated concept to the
anchor concept, will these associated concepts be irrelevant? For example, if you want to unlearn Tobey Maguire, the proposed method may unlearn the character Spider-Man. However, other actors like Andrew Garfield and Tom Holland also played this role. What would happen to other actors?
- The proposed method uses the cosine and/or distance in Euclidian space to find the associated concepts. However, there are no empirical or theoretical results to support this point. Why do you think concepts of Euclidian-close embedding vectors are also close in the semantic space? I would suggest more experiments to verify your claims.
- Minor points, some quotation marks in this manuscript are incorrect, e.g., lines 088, 209. Also, some terms in this manuscript are not consistent, e.g., "concept residue" in the abstract and "concept residual" in the limitation section.
- Will you release the codes for reproduction?

---

### Official Review · Reviewer_ux1N · 2024-11-04

**Soundness:** 2
**Presentation:** 3
**Contribution:** 2
**Rating:** 5
**Confidence:** 4

**Summary:**

This paper proposes a neighbor-concept mining method to address the "concept residue" issue, aiming to expand the erasing range and eliminate the generation of inputs related to the erased concept.

**Strengths:**

1. The paper addresses a significant challenge in concept erasing, where semantically related inputs can evade erasure, limiting the reliability of concept removal.
2. The structure of the paper is well-organized.

**Weaknesses:**

1. The approach to identifying concept residue is fairly simplistic, setting a radius around the neighborhood based on manually selected hyperparameters. This may not effectively capture all related concepts, and it risks inadvertently affecting unrelated concepts.

2. The experimental results are insufficient. Prior research suggests that applying adversarial perturbations [1][2] to the input prompt can reveal erased concepts. Testing the method under adversarial perturbations would be essential to confirm that the concept residue issue has been truly addressed. The experiment detail of how to find the synonym classes is missing. Evaluation metrics on the generation quality of unrelated concepts, such as FID (Fréchet Inception Distance) and CLIP scores, should be included.

[1] Chin Z Y, Jiang C M, Huang C C, et al. Prompting4debugging: Red-teaming text-to-image diffusion models by finding problematic prompts. ICML 2024

[2] Zhang Y, Jia J, Chen X, et al. To generate or not? safety-driven unlearned diffusion models are still easy to generate unsafe images... for now. ECCV 2024

**Questions:**

Why should we first obtain the closed-form solution before refining with LoRA fine-tuning, rather than applying LoRA from the begining?

---

### Official Review · Reviewer_RWik · 2024-11-04

**Soundness:** 3
**Presentation:** 2
**Contribution:** 2
**Rating:** 3
**Confidence:** 4

**Summary:**

In this work, the authors introduce Neighbor-Concept Mining, a novel framework for semantically separating unsafe concepts from safe, retained concepts. The paper proposes two main strategies: Neighbor-Concept Mining and Beyond-Concept Regularization. By leveraging hyperparameters to partition areas within the same manifold, the authors successfully delineate unsafe concept regions from safe retention regions. Concepts embeddings are then sampled from these distinct regions to facilitate the subsequent concept erasure process. This method offers a way to address safety issues in text-to-image model.

**Strengths:**

1.This paper tackles a significant issue in the field of AI safety by focusing on the identification and mitigation of potential generative risks within models. The problem explored in this paper is crucial, especially as generative models become more widely deployed.

2. The authors introduce a method of mining neighboring concepts, examining the relationships between harmful concepts and unrelated concepts. This approach demonstrates a deep understanding of the underlying mechanisms within the concept space and reflects an insightful approach to concept erasure. Additionally, the paper is well-structured with a clear, logical flow that enhances reader comprehension.

**Weaknesses:**

1.The authors claim that preserving unrelated concepts in a larger range in the same manifold space helps preserving the model ability.
However, for nudity erasure, the preservation results on the MS-COCO 30K dataset show limitations. The model's Clip Score is relatively low, indicating its capacity to retain unrelated concepts is still insufficient.

2.The authors propose a method to improve the robustness of concept erasure by sampling and repeatedly erasing concepts neighboring the target concept. This approach bears similarity to methods in recent works such as “Reliable and Efficient Concept Erasure of Text-to-Image Diffusion Models” and “Receler: Reliable Concept Erasing of Text-to-Image Diffusion Models via Lightweight Erasers.” A comparison with these methods would provide valuable insights into the effectiveness and efficiency of the proposed approach.

3. In the nudity erasure experiment, AC appears to perform below the original SD1.4 model, which raises the possibility of inconsistencies in the testing process. Additionally, the results for object erasure using SPM are notably lower than expected. These observations suggest that a more in-depth review of the evaluation setup could be beneficial to confirm the reliability of the reported outcomes and ensure fair comparisons across methods.

4. Adding artistic styles erasure experiments would enhance the comprehensiveness of the study and allow for a broader assessment of the model's erasure capabilities across different types of concepts.

**Questions:**

Please refer to the weakness part.

---

### Note · Authors · 2024-11-15

I have read and agree with the venue's withdrawal policy on behalf of myself and my co-authors.